# Ad Hoc Mesh Network Localization Using Ultra-Wideband for Mobile Robotics

**DOI:** 10.3390/s24041154

**Published:** 2024-02-09

**Authors:** Marius F. R. Juston, William R. Norris

**Affiliations:** Department of Industrial and Enterprise Systems Engineering, University of Illinois Urbana-Champaign, Urbana, IL 61801, USA; wrnorris@illinois.edu

**Keywords:** ultra-wideband (UWB), unscented Kalman filter (UKF), ad hoc localization

## Abstract

This article explores the implementation of high-accuracy GPS-denied ad hoc localization. Little research exists on ad hoc ultra-wideband-enabled localization systems with mobile and stationary nodes. This work aims to demonstrate the localization of bicycle-modeled robots in a non-static environment through a mesh network of mobile, stationary robots, and ultra-wideband sensors. The non-static environment adds a layer of complexity when actors can enter and exit the node’s field of view. The method starts with an initial localization step where each unmanned ground vehicle (UGV) uses the surrounding, available anchors to derive an initial local or, if possible, global position estimate. The initial localization uses a simplified implementation of the iterative multi-iteration ad hoc localization system (AHLos). This estimate was refined using an unscented Kalman filter (UKF) following a constant turn rate and velocity magnitude model (CTRV). The UKF then fuses the robot’s odometry and the range measurements from the Decawave ultra-wideband receivers stationed on the network nodes. Through this position estimation stage, the robot broadcasts its estimated position to its neighbors to help the others further improve their localization estimates and localize themselves. This wave-like cycle of nodes helping to localize each other allows the network to act as a mobile ad hoc localization network.

## 1. Introduction

### 1.1. Overview

Due to the GPS’ constant need for high-accuracy and fidelity in global localization, issues arise when GPS access is limited in some environments. This could happen due to inadequate GPS accuracy or unavailability in indoor or urban environments, such as tunnels [1,2]. Some sensors complement GPS-denied systems, such as LiDARs, camera systems (mono or stereo), inertial measurement sensors, velocity sensors, altimeters, compasses, and more; however, most of the solutions are expensive or prone to more significant errors over time with no way to calibrate [3,4]. Ultra-wideband sensors are ideal for implementing a local GPS because they provide centimeter range accuracy, high range reliability, and low latency [5,6,7]. Despite GPS being affordable and reliable, several issues can make its use suboptimal. Issues include GPS being unavailable in indoor environments and only having five meters of precision, which is not ideal for a higher level of accuracy. This can be most readily noted in a lawn mower example, where 5 m can be a substantial error relative to the robot’s workspace area. Additional issues in a lawn-mower-type application are the potential of trees obstructing the GPS signal and the GPS’s low update rate of 1 Hz, which is insufficient to track fast vehicles. Thus, systems such as UWB, which can provide centimeter-level accuracy at higher rates, are more effective solutions. An RTK solution could be implemented to use centimeter-level accuracy for GPS; however, these are not cost-effective for such situations.

#### Ultra-Wideband Sensors

The use of ultra-wideband (UWB) in localization is a popular technology due to its accurate positioning capabilities, immunity against multipath fading, and resilience against active and passive interferences [5,8]. UWB sensors utilize a large frequency spectrum, from 3.1 to 10.6 GHz, and bandwidths of 500+ MHz to implement localization techniques. This means that UWB radar sensors can, compared to traditional optical or less powerful radar sensors, maintain their accuracy in more challenging terrain, such as indoor situations, where multipath and interference errors are more prevalent. Thus, they retain the ability to sense and communicate in obstacle-heavy terrain and allow for more robust localization systems [9]. There are three node types in UWB sensors: the transmitters, which only transmit their operations; the anchor nodes, which are usually static with a set and verified location to help the other nodes in localizing; and mobile nodes, which contain a combination of transmitting antennas and receiving antennas.

### 1.2. Related Works

UWB localization problems were approached through different configurations of anchors and tags, combining stationary and mobile tags. Two main groups of UWB localization approaches were found in the current literature. The first approach, followed by [10,11,12,13,14,15,16,17], used a combination of initial location estimates using the UWB ranging measurements and a regression method followed by the use of a type of Kalman filter to further improve the estimate over time. The second approach followed a Monte Carlo-based simulation similar to particle filters [18].

To simplify communication issues and improve network robustness, decentralized sensors were most commonly utilized to perform the computation [16]. The decentralized initial location estimates follow a linear regression problem to solve triangulation or trilateration to obtain a robust position estimate [12,17]. However, the issue was that these systems assumed that the robot was constantly able to achieve communication with a minimum number of nodes before adequately operating. The robot was also assumed to be stationary during this period.

Current literature has explored more noise-robust extensions of nonlinear least squares for triangulation. Systems using robust statistical techniques can mitigate the impact of outliers in measurements [19,20,21]. Some employ reformations of the reweighed least squares (IRLS) methods [22], while others transform the LSQ problem using a majorization–minimization (MM) approach [23]. While these methods usually look at static environments, some mobile beacon-based location methods track the anchor’s messages and localize themselves based on the message history [24,25]; however, even in these cases, the tracked nodes are static.

Multiple tags and anchors were placed on the robot to improve the understanding of its position and orientation estimates. This helped constrain its position and provided more reliability in its measurements [15,17]. An issue with previous work was that the systems localized each other in a relative sense; the nodes were localized relative to a base node. This was, however, not ideal for long-distance or multi-mesh scenarios as it did not ensure that each node was constrained to the same coordinate frame.

Once the position estimate was calculated, the system transitioned to position refinement techniques to improve further the time-varying sensor noise provided to the robot. The extended Kalman filter (EKF) and the unscented Kalman filter (UKF) have been used extensively in the literature [15,16]. The UKF is sensitive to the inherent time invariant multipath effect and non-line-of-sight (NLOS) noise. To reduce the inherent error, different types of filter approaches have been used [6,10,14,26]. The noise increase was relatively significant for indoor scenarios because so many obstacles could block the direct line of sight of the UWB sensors. As such, using UWB sensors only for localization was not ideal. The noise inherent in the system was reduced, as demonstrated in this research, with the fusion of the wheel encoders and inertial measurement units (IMUs) to smooth out the position estimate.

In contrast to the aforementioned methods, the second method for position estimation found in the literature followed a Monte Carlo approach [18]. The algorithms followed a repeated random sampling process to arrive at a computationally convergent solution or solutions [27,28]. This sampling technique does provide possible issues with multiple equilibrium points and can be more computationally expensive than other options, depending on the number of sampling points. Efforts to enhance wireless network localization techniques have delved into integrating machine learning within the localization engine [29]. Semi-supervised particle swarms have been augmented to improve the data point selection; however, currently, these systems still assume that the anchor nodes are static [28]. The following section introduces this research problem formulation.

### 1.3. Contribution

This article focused on the localization of moving unmanned ground vehicles (UGV) using stationary and moving UGV and unmanned aerial vehicles (UAV). These vehicles had UWB sensors attached, enabling peer-to-peer ranging communication and allowing the robots to improve their respective localization using a cooperative network. The network helped provide a broad reach and create a local GPS network for the nearby robots.

The main contributions from the research are highlighted below:1.Developed an initial localization framework where all agents could be non-stationary, and the UWB tags are offset from the center of the robot. This novel initial localization enables a fully mobile global localization system, where the localizing nodes do not need to maintain a stationary position to derive initial position estimates in their environment. This method also reduces the number of required nodes to fully localize as compared to a static environment.2.Designed a global/relative localization system based on the UKF for ground-based robots that could leverage ground and aerial range measurement data. This capability allows for a large variety of input sources for improved accuracy in the tracking of ground-based robots.3.Created a pipeline for a mobile ad hoc localization system. This wave-like self-expanding mesh network of UWB nodes enables the deployment and maintenance of a large mobile localization engine.

The rest of the article was organized as follows. The environment and problem are described in Section 2, where the agents of the environment, the assumptions of the environment, and the detailed derivations of the proposed system are laid out. Section 3 contains the simulation results and interpretation. Finally, the work summary is presented in Section 4.

## 2. Method

The ad hoc mesh network definition and implementation, including the explanation of the information propagation throughout the network, are explained in the following subsection. The remaining sections derive the initial position transform through a nonlinear least squares regression, starting in Section 2.3 and ending in Section 2.4, followed by the position refinement using an unscented Kalman filter, starting in Section 2.5 and ending in Section 2.5.4.

### 2.1. Ad Hoc Network

The proposed ad hoc mesh network followed a mobile decentralized, absolute localization ad hoc system. The nodes in the network would consist of both stationary and mobile nodes. Due to the network’s mobile nature, a decentralized network was more robust to the constantly changing topography of the network graph. The network was fine-grained (range-based) thanks to the incorporated sensors in this research, the UWB sensors, measuring the range using ToF (time of flight). The system was also required to converge to an absolute localization system; this means that the robot should be able to transition from an initial relative localization system, and once it had reached the required threshold for transitioning, convert the relative coordinate system to become a global localization problem.

The algorithm follows a similar but simplified version of the iterative multilateration AHLoS (ad hoc localization system)  [30]. The AHLoS is a method to implement the localization of a mesh network in a flood-like fashion. The algorithm started with a graph that combined localized and unlocalized nodes; when an unknown node lies in the neighborhood of three or more anchors, the neighboring anchors’ positions and distances were used to estimate its position. Once the position of an unknown node was estimated, the node became an anchor and thus continued the cycle until all the nodes were localized.

This article modified the AHLoS system to account for mobile nodes coming out and into the range of other localized nodes. The current and historical range measurements were only collected from the localized anchors and their associated position at every time step. The range measurements were then paired with the node’s odometry measurement at the corresponding time. Linear extrapolation was employed if the odometry measurements were too far apart.

The localized nodes communicate to the target robot the range measurement and the localized node’s current global position. Given the distributed nature of the mobile ad hoc network, each robot would maintain a history of measurement data from each robot. With nodes frequently entering and exiting the robot’s view and with histories stored locally, the system retains all information and remains unaffected by network topography changes. Given that the range measurement uses ToF for its UWB range estimation, no synchronization between nodes is required, thus making communication more straightforward. As for transmission scheduling, the system would adopt a carrier-sense multiple access with collision avoidance (CSMA/CA) protocol to communicate with the other devices [31]. The device first listens to the UWB channel to check for transmitting nodes. If another node is detected, the device waits a random interval for that node to finish before checking again and then sending its data. This technique is known to be unreliable due to the hidden node problem; however, given UWB’s low transmission rate, the UWB range packets being of small size, and the expected sparsity of the robot at a specific time (no more than ten robots at a time), the risk for package collision is low. The CSMA/CA would be able to handle such traffic with acceptable system delay [32]. If, after sending, packets are not returned within a time frame T, the ranging handshake would be sent again.

Following the data collection, several data processing checks were employed before following through with the initial localization step:1.If the current robot was stationary, then a minimum of three unique anchors were required;2.If the current robot was non-stationary and the anchors were stationary, then a minimum of two anchors were required;3.If the current robot was non-stationary and the anchors were non-stationary, then a minimum of one anchor was required.

The anchors were determined to be mobile by looking at the reported robot’s positions and ensuring the distances between points were above a certain tolerance. This provided enough information for the nonlinear least square algorithm to work.

To mitigate issues with sensor noise, a minimum number of data points was enforced to start the trilateration. Once the quota was fulfilled, the nonlinear least square algorithm was performed as described in Section 2.4, the robot was defined as localized, and the refining process using the UKF was started. However, if the node failed to localize within a specific time frame, the robot relied solely on odometry for the localization process and “positioned itself within its relative positioning system”. Suppose the node was at any time able to satisfy all the conditions to localize due to it coming in range with other localized nodes. In that case, the robot switched from using the relative positioning system to localize itself in the new global reference frame and continued using the UKF to refine the results. At this point, the robot switched from an unlocalized node to a localized node. By constantly localizing and changing status once localized, the robots eventually converged to a network where all the nodes followed the same reference frame.

### 2.2. Parameter Definitions

A detailed definition of all parameters used in the system will be presented in this subsection. For convention, the vectors p and χ, ϕ represented the global position, relative position, and heading of the node, respectively. The heading was defined where 0 radians points in the positive *x*-axis of p, and a counterclockwise motion represented an increase in the heading. χ represented the inter distance between two points in an agreed reference frame and can be formulated as,
(1)$$\chi_{\mathrm{ij}}=\vec{P_{i}P_{j}}=p_{j}-p_{i}=\left(x_{j}-x_{i},y_{j}-y_{i},z_{j}-z_{i}\right)$$

The relative Euclidean distance between two nodes was thus denoted as
(2)$$\|\chi_{\mathrm{ij}}\|=r_{ij}=\sqrt{{\left(x_{j}-x_{i}\right)}^{2}+{\left(y_{j}-y_{i}\right)}^{2}+{\left(z_{j}-z_{i}\right)}^{2}}$$

The environment consisted of a team of UAV, UGV, and static UWB anchors labeled 1, 2, ..., N. The calculated global transformation matrices of the UGV were defined as the transformation from the established global origin point to the UGV’s odometry origin point in the global reference frame. The origin point of the odometry was assumed to be transformed by (0,0,0), with 0 heading, linear and angular velocity, to the robots’ initial global position ($\chi_{0,0}$). The transformation thus translated and rotated the static odometry frame of the robot. To solve the problem, each $UGV_{i}$ was able to access the odometry data, denoted as $\left(\chi_{x_{i},0},\chi_{y_{i},0},\chi_{z_{i},0}\right)$ with angular velocity $\omega_{i}$, linear velocity $v_{i}$, heading $\Delta \theta_{i}$, also represented as $\delta \theta =\arctan\left(\frac{v_{y}}{v_{x}}\right)$, and $\dot{\theta }$ represented the yaw rate. The robot had distance measurements to its neighbor $UGV_{j}$, i.e., $r_{ij}=\|\chi_{\mathrm{ij}}\|$. It was assumed that the robot was moving on a flat 2D plane. As such, ${\hat{p}}_{z}$ remained constant. The global position p was then derived from $\left(x_{0},y_{0},z_{0}\right),\theta_{0}$ by:(3)$$\begin{array}{c} p=\left[\begin{array}{ccccc} \cos\theta_{0} & -\sin\theta_{0} & 0 & 0 & x_{0}\\ \sin\theta_{0} & \cos\theta_{0} & 0 & 0 & y_{0}\\ 0 & 0 & 1 & 0 & z_{0}\\ 0 & 0 & 0 & 1 & \theta_{0} \end{array}\right]\left[\begin{array}{c} \chi_{x,i}\\ \chi_{y,i}\\ \chi_{z,i}\\ \Delta \theta_{i}\\ 1 \end{array}\right] \end{array}$$

Each $UGV_{i}$ denoted the set of its neighbors as $UGV_{j}$ where {j∣j∈Zandi≠j}. The neighbor transmitted its range measurements and UWB anchor position, $A_{k}$. The position measurement data were located in the robot’s global reference frame, i.e., $p_{j}=\left(x_{j},y_{j},z_{j}\right)$. Each UWB anchor was annotated as $A_{k}$ with a unique identification *k* where {k∣k∈Z}, and similarly, each UWB tag was denoted as $T_{w}$ with a unique identification *w*, where {w∣w∈Z}.

### 2.3. Initial Localization

An initial position estimate was made on the target node. The proposed algorithm solved this problem by structuring the estimate as a nonlinear least square (NLS) problem. To simplify the construction of the NLS, the odometry and neighboring range data of a single robot node were used to calculate $\left(x_{0},y_{0},z_{0}\right),\theta_{0}$. The position estimate was later refined using the UKF.

The geometry of the robot is defined and elaborated on below. There were two tags on the robot, which were separated by a fixed distance *d*. In the global reference frame, the tags were defined as $T_{w}$. In the static inertial reference frame of the robot, the position of the tags was represented as $t_{w}$, where the left tag was at $t_{l}=\left(0,\frac{d}{2}\right)$, and the right tag was at $t_{r}=\left(0,-\frac{d}{2}\right)$. In addition, the odometry position was measured from the robot’s defined origin, (0,0), which was the midpoint of the right and left tags. $p_{i}=\frac{T_{l,w1}+T_{r,w2}}{2}$, and therefore, $\left(0,0\right)=\frac{t_{l}+t_{r}}{2}$. The range recorded by the UWB tags was represented as $\left\|A_{j}-T_{w}\right\|=d_{jw}$, where $A_{j}$ was a UWB anchor located on a UGV neighbor.

The robot’s heading $\phi_{i}$ was considered when solving the nonlinear least squares problem by compensating the target distance by rotating the tag pose. The relative tag position, defined as $\left(T_{w}-p_{i}\right)$, was represented as the vector $t_{w}$ rotated about the center of the robot ($p_{i}$) by the heading global heading $\phi_{i}$.

### 2.4. Nonlinear Least Squares

The nonlinear least squares optimization method seeks to minimize the following:(4)$$S=\sum_{i=1}^{m}r_{i}^{2}$$
where $r_{i}$ was the residual given by:(5)$$r_{i}=d_{jw,calculated}^{2}-d_{jw,measured}^{2}$$

The range measurement $d_{jw}$ was defined as a function of $p_{i},\phi_{i},t_{w}\;\mathrm{and}\;A_{j}$. In addition, $\Delta \theta_{i}$ represented the current heading measured from the odometry and $\chi_{i}$ represented the odometry position $\left(\chi_{x,i},\chi_{y,i},\chi_{z,i}\right)$: (6)$$\begin{array}{cc} d_{jw} & =\left\|\vec{A_{j}}-\vec{T_{w}}\right\|\\ \; & =\left\|\vec{A_{j}}-\left(\vec{p_{i}}+\left[\begin{array}{ccc} \cos\phi_{i} & -\sin\phi_{i} & 0\\ \sin\phi_{i} & \cos\phi_{i} & 0\\ 0 & 0 & 1 \end{array}\right]\vec{t_{j}}\right)\right\|\\ \; & =∥\vec{A_{j}}-(\left(\vec{p_{0}}+\left[\begin{array}{ccc} \cos\theta_{0} & -\sin\theta_{0} & 0\\ \sin\theta_{0} & \cos\theta_{0} & 0\\ 0 & 0 & 1 \end{array}\right]\chi_{i}\right)+\\ & \left[\begin{array}{ccc} \cos\left(\theta_{0}+\Delta \theta_{i}\right) & -\sin\left(\theta_{0}+\Delta \theta_{i}\right) & 0\\ \sin\left(\theta_{0}+\Delta \theta_{i}\right) & \cos\left(\theta_{0}+\Delta \theta_{i}\right) & 0\\ 0 & 0 & 1 \end{array}\right]\vec{t_{w}})∥ \end{array}$$
(7)$$\begin{array}{cc} d_{jw}^{2} & =(A_{jx}-(x_{0}+\left[\chi_{x,i}\cos\theta_{0}-\chi_{y,i}\sin\theta_{0}\right]+\\ \; & \left[t_{wx}\cos\left(\theta_{0}+\Delta \theta_{i}\right)-t_{wy}\sin\left(\theta_{0}+\Delta \theta_{i}\right)\right]{))}^{2}+\\ \; & (A_{jy}-(y_{0}+\left[\chi_{x,i}\sin\theta_{0}+\chi_{y,i}\cos\theta_{0}\right]+\\ \; & \left[t_{wx}\sin\left(\theta_{0}+\Delta \theta_{i}\right)+t_{wy}\cos\left(\theta_{0}+\Delta \theta_{i}\right)\right]{))}^{2}+\\ \; & {\left(A_{jz}-\left(z_{0}+t_{wz}\right)\right)}^{2} \end{array}$$

The minimum value of *S* could be found when the gradient was 0. As a result, a Jacobian was used to calculate the partial derivative of all the *n* variables needed, where $\beta_{j}$ represented the optimization parameters that were optimized—in this case, $x_{0},y_{0},z_{0},\theta_{0}$.
(8)$$\begin{array}{ccc} \frac{\delta S}{\delta \beta_{j}} & =2\sum_{i=1}^{m}r_{i}\frac{\delta r_{i}}{\delta \beta_{j}}=0 & \left(j=1,\cdots ,n\right) \end{array}$$

Due to the gradient equations not having a closed solution, the variables $\beta_{j}$ were solved iteratively with Newton’s iteration method.
(9)$$f\left(x_{i},\beta \right)\approx f\left(x_{i},\beta^{k}\right)+\sum_{j}J_{ij}\Delta \beta_{j}$$

The Jacobian is a matrix of partial derivatives as a function of constants, the independent variable, and the parameter. It was constructed as such:(10)$$\frac{\delta r_{i}}{\delta \beta_{j}}=J_{ij}$$

A single row of the Jacobian ( *J* ) was derived from the $d_{iw}^{2}$ function below. In this matrix c,s represent cos(·) and sin(·), respectively.
(11)$$J^{T}=\left[\begin{array}{c} \frac{\delta d_{jw}^{2}}{\delta x_{0}}\\ \frac{\delta d_{jw}^{2}}{\delta y_{0}}\\ \frac{\delta d_{jw}^{2}}{\delta z_{0}}\\ \frac{\delta d_{jw}^{2}}{\delta \theta_{0}} \end{array}\right]=\left[\begin{array}{c} -2(A_{x}-t_{x}c\left(\theta_{0}+\Delta \theta_{i}\right)+t_{y}s\left(\theta_{0}+\Delta \theta_{i}\right)\\ -c\left(\theta_{0}\right)\chi_{x,i}-x_{0}+s\left(\theta_{0}\right)\chi_{y,i})\\ 2(-A_{y}+t_{x}s\left(\theta_{0}+\Delta \theta_{i}\right)+t_{y}c\left(\theta_{0}+\Delta \theta_{i}\right)\\ +s\left(\theta_{0}\right)\chi_{x,i}+c\left(\theta_{0}\right)\chi_{y,i}+y_{0})\\ 2(-A_{z}+t_{z}+z_{0})\\ 2(t_{x}c\left(\theta_{0}+\Delta \theta_{i}\right)-t_{y}s\left(\theta_{0}+\Delta \theta_{i}\right)+\\ c\left(\theta_{0}\right)\chi_{x,i}-s\left(\theta_{0}\right)\chi_{y,i})\\ (-A_{y}+t_{x}s\left(\theta_{0}+\Delta \theta_{i}\right)+t_{y}c\left(\theta_{0}+\Delta \theta_{i}\right)+\\ s\left(\theta_{0}\right)\chi_{x,i}+c\left(\theta_{0}\right)\chi_{y,i}+y_{0})\\ -2(t_{x}s\left(\theta_{0}+\Delta \theta_{i}\right)+t_{y}c\left(\theta_{0}+\Delta \theta_{i}\right)+\\ s\left(\theta_{0}\right)\chi_{x,i}+c\left(\theta_{0}\right)\chi_{y,i})\\ (-A_{x}+t_{x}c\left(\theta_{0}+\Delta \theta_{i}\right)-t_{y}s\left(\theta_{0}+\Delta \theta_{i}\right)+\\ c\left(\theta_{0}\right)\chi_{x,i}+x_{0}-s\left(\theta_{0}\right)\chi_{y,i}) \end{array}\right]$$

Using Newton’s iteration algorithm for minimizing the error, the Jacobian in Equation (Equation 11) and the residual functions $r_{i}$ were calculated at each step. The residual function was redefined as $f_{i}=d_{jw,i}{\left(x_{0},y_{0},z_{0},\theta_{0}\right)}^{2}-d_{jw,i,measured}^{2}$, and the vectors f and β were introduced as:(12)$$\begin{array}{c} J=2\left[\begin{array}{cccc} \frac{\delta f_{1}}{\delta x_{0}} & \frac{\delta f_{1}}{\delta y_{0}} & \frac{\delta f_{1}}{\delta z_{0}} & \frac{\delta f_{1}}{\delta \theta_{0}}\\ \frac{\delta f_{2}}{\delta x_{0}} & \frac{\delta f_{2}}{\delta y_{0}} & \frac{\delta f_{2}}{\delta z_{0}} & \frac{\delta f_{2}}{\delta \theta_{0}}\\ \vdots & \vdots & \vdots & \vdots \\ \frac{\delta f_{n}}{\delta x_{0}} & \frac{\delta f_{n}}{\delta y_{0}} & \frac{\delta f_{n}}{\delta z_{0}} & \frac{\delta f_{n}}{\delta \theta_{0}} \end{array}\right],f=\left[\begin{array}{c} f_{1}\\ f_{2}\\ \vdots \\ f_{n} \end{array}\right],\beta =\left[\begin{array}{c} x_{0}\\ y_{0}\\ z_{0}\\ \theta_{0} \end{array}\right] \end{array}$$

Thus, the Newton iteration became:(13)$$\beta_{k+1}=\beta_{k}-{\left(J_{k}^{T}J_{k}\right)}^{-1}J_{k}^{T}f_{k}$$
where $\beta_{k+1}$ represented the new estimated parameters, and $\beta_{k}$ was the last approximation. The initial estimate needed for the Newton approximation ($\beta_{0}$) was provided in the simulation. The parameter $\beta_{0}$ can be initialized in one of two ways: it can either be set to an initial state of 0, or it can be provided as a coarse estimate of the robot’s geographic area. The algorithm terminated when $|\beta_{k+1}-\beta_{k}|\leq \epsilon$, where ϵ was a defined a termination tolerance criterion.

This was this paper’s approach to solving the nonlinear least square optimization. Additional extensions to the NLLS could be used to improve the solution depending on the particular problem. The scipy.optimize.least_squares function was used in this work to implement the nonlinear least square calculations.

#### Monte Carlo Simulation

A Monte Carlo simulation was developed to demonstrate the convergence characteristics of the initial localization process. This simulation involved the manipulation of two critical variables: the number of robots traversing the environment, ‘N’, and the time horizon for the odometry history, ‘T’. In the simulation, ‘N’ random robots were created, each adhering to a differential drive motion model. At each time step (0.1 s), these simulated robots moved with a random linear velocity selected from the range of [−vmax/2, vmax], favoring forward motion, and an angular velocity sampled from [−wmax, wmax], where both vmax and wmax were set to 2. The control inputs were sampled uniformly within their respective ranges. Simultaneously, one robot was randomly selected to transmit its range measurements at each time step.

The parameter ‘T’ governed the duration before the localization process was conducted using the collected data. Random noise was introduced to the acquired odometry data, affecting both anchor poses and the target robot’s measurements. The noise levels were standardized using a zero mean Gaussian distribution with standard deviations set to 0.5 cm for the x and y coordinates of the odometry anchor poses, 2 cm for the range measurements, and 1/1000 radians for angular measurements. These noise standard deviations were deemed appropriate given the sensors available for the following reasons. Assuming that the anchor poses were using an RTK or a similarly high-accuracy localization engine for their positioning, a standard deviation of 0.5 cm was deemed appropriate. The Decawave DWM1000 Modules were observed to have an approximately 2 cm deviation in their range measurements, and onboard Microstrain IMU was estimated to have an approximate noise of 1/1000 radians. The target robot was uniformly randomly placed within a 10 × 10 m world with a randomly assigned orientation, as demonstrated in Figure 1. Additionally, the nonlinear least squares’ initial guesses were set to zero.

Each parameter combination was simulated with 1000 samples to collect the aggregated information. These simulations generated values of ‘T’ from 1 s to 10 s and an ‘N’ ranging from 1 to 5 robots. The results, as depicted in Figure 2 and Figure 3, indicated that as the time horizon ‘T’ increased, the Euclidean error relative to the target transformation matrix diminished significantly. In the figures below, the median error was used to demonstrate the error over the runs with the light-shaded regions representing the interquartile range of the error. Furthermore, a higher number of robots present led to less variation in the error relative to the target across runs, and the system was less likely to reach an incorrect solution.

Figure 4 and Figure 5 demonstrate the robustness of the system relative to noise levels. The Monte Carlo simulations were generated again; however, this time, assuming a constant ‘T’ of 10 s and a varying scale of the noise levels listed above at different magnitudes.

When N = 2, the maximum error is often larger than at N = 1. This may be due to the fact that when optimizing with N = 2, it is more likely to reach a local minimum along an approximate axis of symmetry due to the angle’s nonlinearity. This causes an invalid solution to be returned. Further research in using quaternions instead of Euler angles when optimizing, which follows a more linear mapping, could help reduce the nonlinearities in the system. This would allow for different, more robust optimization methods.

### 2.5. UKF Position Refinement

In the following sections, the robot has already been localized and given an initial position estimation. Further refinement of the results by reducing the sensor noise was achieved using a UKF [33,34]. Two types of data were used in the UKF. The first came from the ranging measurements at time step *k*, $d_{k}^{jw}$, between the tag $T_{w}$ on the target node and a localized neighbor anchor $A_{j}$. The second piece of data came from the robot’s onboard odometry. The following sections go through the UKF model for the CTRV motion model.

#### 2.5.1. Motion Model

The UKF motion model that this article followed was the constant turn rate and velocity magnitude model (CTRV). This nonlinear motion model assumed that the node could move straight but turned following a bicycle turn model with constant turn rate and linear velocity [35]. Other types of motion models were available, as illustrated in Figure 6. The constant velocity (CV) motion model was the simplest one available, a linear motion model where the linear acceleration was defined to be 0. The constant turn rate and acceleration (CTRA) is an expanded version of the CTRV motion model where the acceleration is accounted for and determined. Similarly to the CTRA motion model, the constant curvature and acceleration (CCA) model replaces the yaw rate of the model with the curvature instead. The constant steering angle and velocity (CSAV) model returns to having the acceleration constant and replacing the assumed constant steering angle instead.

Each model has its own set of advantages and disadvantages; however, the CTRV motion model was chosen thanks to the balance in computation speed and accuracy in comparison with each different model [35]. The CTRV was also selected because the odometry sensor output contained the same state variables. This research problem formulation assumes the nodes of interest are ground robots; however, a general motion model could be used. As long as the robot’s dynamics can be derived, the system’s motion model can be used with this implementation. For drones, the motion model is linear. This eliminates the need for an unscented Kalman filter model, allowing the use of a linear Kalman filter instead. The state vector of the CTRV model was defined to be (Figure 7):(14)$$x={\left[\begin{array}{cccccc} p_{x} & p_{y} & p_{z} & \upsilon & \psi & \dot{\psi } \end{array}\right]}^{T}$$
where υ was the node’s speed, ψ was the yaw angle, which described the orientation according to Figure 7, and $\dot{\psi }$ represented the yaw rate. The change in rate in the state was expressed as:(15)$$\;\begin{array}{cc} \dot{x} & ={\left[\begin{array}{cccccc} \dot{p_{x}} & \dot{p_{y}} & \dot{p_{z}} & \dot{\upsilon } & \dot{\psi } & \ddot{\psi } \end{array}\right]}^{T}\\ \; & ={\left[\begin{array}{cccccc} \upsilon \cdot \cos\psi & \upsilon \cdot \sin\phi & 0 & 0 & \dot{\psi } & 0 \end{array}\right]}^{T} \end{array}$$

The current step in the process was denoted by *k* and the subsequent step by k+1. Consequently, the time difference was expressed as $\Delta t=t_{k+1}-t_{k}$. The process model, which predicted the state at k+1, could be decomposed into the deterministic and stochastic parts. To derive the deterministic part:(16)$$\begin{array}{cc} x_{k+1} & =f_{2}\left(x_{k}\right)\\ \; & =x_{k}+\int_{t_{k}}^{t_{k+1}}{\left[\begin{array}{cccccc} \upsilon \cdot \cos\psi (t) & \upsilon \cdot \sin\phi (t) & 0 & 0 & \dot{\psi } & 0 \end{array}\right]}^{T}dt\\ \; & =x_{k}+\left[\begin{array}{c} \upsilon_{k}\int_{t_{k}}^{t_{k+1}}\cos\psi (t)dt\\ \upsilon_{k}\int_{t_{k}}^{t_{k+1}}\sin\phi (t)dt\\ 0\\ 0\\ \dot{\psi }\Delta t\\ 0 \end{array}\right]\\ & =x_{k}+\left[\begin{array}{c} \upsilon_{k}\int_{t_{k}}^{t_{k+1}}\cos\left(\psi_{k}+\dot{\phi_{k}}*\left(t-t_{k}\right)\right)dt\\ \upsilon_{k}\int_{t_{k}}^{t_{k+1}}\sin\left(\psi_{k}+\dot{\phi_{k}}*\left(t-t_{k}\right)\right)dt\\ 0\\ 0\\ \dot{\psi }\Delta t\\ 0 \end{array}\right]\\ \; & =x_{k}+\left[\begin{array}{c} \frac{\upsilon_{k}}{\dot{\psi_{k}}}\left(\sin\left(\psi_{k}+\dot{\psi_{k}}\Delta t\right)-\sin\psi_{k}\right)\\ \frac{\upsilon_{k}}{\dot{\psi_{k}}}\left(-\cos\left(\psi_{k}+\dot{\psi_{k}}\Delta t\right)+\cos\psi_{k}\right)\\ 0\\ 0\\ \dot{\psi }\Delta t\\ 0 \end{array}\right] \end{array}$$

However, problems arose when $|\dot{\psi }|\approx 0$ as this would cause a division by 0. A modified version of the motion model was thus defined for when $|\dot{\psi }|\leq \varepsilon$:(17)$$f_{2}\left(x_{k}\right)=x_{k}+{\left[\begin{array}{cccccc} \upsilon_{k}\cos\psi_{k}\Delta t & \upsilon_{k}\sin\psi_{k}\Delta t & 0 & 0 & \dot{\psi }\Delta t & 0 \end{array}\right]}^{T}$$

Next, the stochastic component was designated as the noise vector, encompassing the linear and yaw acceleration noises in a CTRV model. At time step *k*, the noise ν was characterized as follows:(18)$$\nu_{k}={\left[\begin{array}{cc} \nu_{a,k} & \nu_{\ddot{\psi },k} \end{array}\right]}^{T}$$
where $\nu_{a,k}$ was the linear acceleration noise defined as a normal distribution, $\nu_{a,k}∼N\left(0,\sigma_{a}^{2}\right)$ and $\nu_{\ddot{\psi },k}$ were the yaw acceleration noise defined as a normal distribution, $\nu_{\ddot{\psi },k}∼N\left(0,\sigma_{\ddot{\psi }}^{2}\right)$. It was assumed that the linear and angular acceleration would remain relatively constant during small time intervals, resulting in approximately linear motion between two timesteps (this assumption was valid unless the yaw rate was excessively high). As a result, the noise function was expressed as follows:(19)$$f_{2}\left(\nu_{k}\right)=\left[\begin{array}{c} \frac{1}{2}{\left(\Delta t\right)}^{2}\cos\psi_{k}\cdot \nu_{a,k}\\ \frac{1}{2}{\left(\Delta t\right)}^{2}\sin\psi_{k}\cdot \nu_{a,k}\\ 0\\ \Delta t\cdot \nu_{a,k}\\ \frac{1}{2}{\left(\Delta t\right)}^{2}\cdot \nu_{\ddot{\psi },k}\\ \Delta t\cdot \nu_{\ddot{\psi },k} \end{array}\right]$$

The full motion model was characterized as follows:(20)$$f\left(x_{k},\nu_{k}\right)=f_{1}\left(x_{k}\right)+f_{2}\left(\nu_{k}\right)$$

#### 2.5.2. State Prediction

Sigma Points

The first step to the unscented Kalman filter was the sigma point generation. Using sigma points is the main difference between the EKF and the UKF. The EKF linearizes the system through a Taylor-series expansion around the mean of the relevant Gaussian random variable (RV) [34]. Using multiple points to sample the state distribution improved the linearization of the nonlinear space [34]. When greater accuracy was required, a UKF was recommended compared to an EKF. Due to the addition and the linear scaling of the number of sigma points required based on the dimensionality of the state vector, UKFs are known to be more computationally expensive. Following a Gaussian distribution, the sigma points were generated from the last updated state and covariance matrix.

First, the augmented state and covariance matrix were formulated, with the normal state vector denoted by:(21)$$x_{k}={\left[\begin{array}{cccccc} p_{x,k} & p_{y,k} & p_{z,k} & \upsilon_{k} & \psi_{k} & \dot{\psi_{k}} \end{array}\right]}^{T}$$

Having a dimension of $n_{x}=6$, the covariance matrix $P_{k|k}$ took the form of an $n_{x}\times n_{x}$ matrix. Meanwhile, the noise vector was characterized as:(22)$$\nu_{k}={\left[\begin{array}{cc} \nu_{a,k} & \nu_{\ddot{\psi },k} \end{array}\right]}^{T}$$
with dimension $n_{\nu }=2$, the augmented state matrix was represented as follows:(23)$$\begin{array}{cc} x_{aug} & ={\left[\begin{array}{c} x|\nu_{k} \end{array}\right]}^{T} \end{array}$$(24)$$\begin{array}{cc} & ={\left[\begin{array}{cccccccc} p_{x,k} & p_{y,k} & p_{z,k} & \upsilon_{k} & \psi_{k} & \dot{\psi_{k}} & \nu_{a,k} & \nu_{\ddot{\psi },k} \end{array}\right]}^{T} \end{array}$$

The augmented state dimension, determined as $n_{a}=n_{x}+n_{\nu }$, yielded a total of eight dimensions for the CTRV model, specifically $n_{a}=8$. Subsequently, the process noise covariance matrix was constructed, incorporating the expected acceleration and yaw rate Gaussian distribution into the matrix *Q*, resulting in the following representation:(25)$$Q=E\left\{\nu_{k}\cdot \nu_{k}^{T}\right\}=\mathrm{diag}\left(\left[\begin{array}{cc} \sigma_{a}^{2} & \sigma_{\ddot{\psi }}^{2} \end{array}\right]\right)$$

The augmented covariance matrix was thus represented as:(26)$$P_{a,k|k}=\mathrm{diag}\left(\left[\begin{array}{cc} P_{k|k} & Q \end{array}\right]\right)$$

The augmented covariance matrix became a square matrix with size $n_{a}\times n_{a}=8\times 8$. By convention, it was recommended to have at least $n_{\sigma }=2n_{a}+1$ sigma points; in the case of the illustrated model, this would be $n_{\sigma }=2\left(8\right)+1=17$. The sigma points were then calculated as a list
(27)$$X_{k|k}=\left[\begin{array}{ccc} x_{k|k} & x_{k|k}+\sqrt{\left(\lambda +n_{a}\right)P_{k|k}} & x_{k|k}-\sqrt{\left(\lambda +n_{a}\right)P_{k|k}} \end{array}\right]$$
where λ was the scaling factor defined as $\lambda =\alpha^{2}\left(n_{x}+3-n_{a}\right)-n_{x}$, where λ dictated how far away from the mean the sigma points would be positioned [37]. The parameter α defined the spread of the sigma points around $x_{k}$ and was set to a small positive value between 0 and 1. By default, it was set to 1; however, this could be tuned if necessary. When taking $\sqrt{P_{k|k}}$ for the sigma points, there were multiple ways to take the square root of a matrix; however, Cholesky decomposition was recommended due to its computation speed. The output of this list should be in the format $n_{a}\times n_{\sigma }$.
Prediction Step

Once the sigma points were calculated and transformed into their respective predicted state, each was transformed using the process model and then saved to another list now of format $n_{x}\times n_{\sigma }$. This process is demonstrated in Algorithm 1. The processed state sigma points were in the format $x_{k+1|k}={\left[\begin{array}{cccccc} p_{x,k+1} & p_{y,k+1} & p_{z,k+1} & \upsilon_{k+1} & \psi_{k+1} & {\dot{\psi }}_{k+1} \end{array}\right]}^{T}$.
**Algorithm 1** Sigma point transformation**Require:** $X_{a,k|k}$, the sigma points of the augmented**Require:** $f(x_{k},\nu_{k})$, the process model function $X_{k+1|k}=\emptyset$ **for all** $x_{a,k|k,i}\in X_{a,k|k}$ **do**    $x_{x,i}\leftarrow x_{a,k|k,i}\left[0:n_{x}\right]$    $x_{\nu ,i}\leftarrow x_{a,k|k,i}\left[n_{x}:n_{a}\right]$    $x_{k+1|k,i}\leftarrow f\left(x_{x,i},x_{\nu ,i}\right)$    Append $x_{k+1|k,i}$ to $X_{k+1|k}$ **end for** **return** 
$X_{k+1|k}$

Calculate Mean and Covariance from the Predicted Points

In this step, the predicted state mean vector and the predicted covariance matrix were calculated by aggregating the sigma points using a weighted average of the points. There were multiple ways to initialize the weights for this part; however, this paper sticks to a standard Gaussian distribution.
(28)$$\begin{array}{cccc} \omega_{i}^{m} & =\frac{\lambda }{\left(\lambda +n_{a}\right)} & \; & i=0 \end{array}$$
(29)$$\begin{array}{cccc} \omega_{i}^{c} & =\frac{\lambda }{\left(\lambda +n_{a}\right)}+\left(1-\alpha^{2}+\beta \right) & \; & i=0 \end{array}$$
(30)$$\begin{array}{cccc} \omega_{i}=\omega_{i}^{m}=\omega_{i}^{c} & =\frac{1}{2\left(\lambda +n_{a}\right)} & \; & i=1,\cdots ,2*n_{\sigma } \end{array}$$
where λ and α were the same from the sigma point calculation in Section 2.5.2. β was an extra parameter used to incorporate any prior knowledge of the distribution of the state. In most cases, this paper’s β was left to be 0. Once the weights were calculated, the predicted mean and covariance were calculated, respectively:(31)$$\begin{array}{cc} x_{k+1|k} & =\sum_{i=0}^{2n_{\sigma }}w_{i}^{m}X_{k+1|k,i} \end{array}$$(32)$$\begin{array}{cc} P_{k+1|k} & =\sum_{i=0}^{2n_{\sigma }}w_{i}^{c}\left(X_{k+1|k,i}-x_{k_{k}+1|k}\right){\left(X_{k+1|k,i}-x_{k_{k}+1|k}\right)}^{T} \end{array}$$

#### 2.5.3. Measurement Prediction

The measurement stage processes the sigma points into predicted measurement outputs. To accomplish this, a measurement function $h(x_{k+1})$ was developed for each of the sensor types, where $x_{k+1}$ was the output of the process model in the prediction step in Section 2.5.2. The measurement vector was $z_{k+1}$, and $\omega_{k+1}$ was the inherent measurement noise.
(33)$$\begin{array}{c} x_{k+1}=f\left(x_{k},\nu_{k}\right) \end{array}$$
(34)$$\begin{array}{c} z_{k+1}=h\left(x_{k+1}\right)+\omega_{k+1} \end{array}$$

Due to this paper tackling the fusion of two sensor types, the odometry data and the UWB range data, two different $h(x_{k+1})$ functions needed to be defined. For the odometry data, the state vector was defined as:(35)$$z_{k+1|k}=h\left(x_{k+1}\right)={\left[\begin{array}{cccccc} p_{x,k+1} & p_{y,k+1} & p_{z,k+1} & \upsilon_{k+1} & \psi_{k+1} & {\dot{\psi }}_{k+1} \end{array}\right]}^{T}$$

The measurement transformation function for the UWB ranging sensors took a similar format to the nonlinear least square $d_{jw}$ derivation (Section 2.4) where the predicted UWB range measurement was derived given the estimated sigma point state ($x_{k+1}$), the known anchor point position ($A_{j}$), and the tag position relative to the node’s reference frame ($t_{w}$).
(36)$$\begin{array}{cc} z_{k+1|k} & =h\left(x_{k+1},A_{j},t_{w}\right)\\ \; & =\left\|\vec{A_{j}}-\vec{T_{w}}\right\|\\ \; & =\left\|\vec{A_{j}}-\left(\vec{p_{k+1}}+\left[\begin{array}{ccc} \cos\psi_{k+1} & -\sin\psi_{k+1} & 0\\ \sin\psi_{k+1} & \cos\psi_{k+1} & 0\\ 0 & 0 & 1 \end{array}\right]\vec{t_{j}}\right)\right\|\\ \; & =\sqrt{\begin{array}{c} {\left(A_{x,j}-\left(p_{x,k+1}+\left[t_{x,w}\cos\psi_{k+1}-t_{y,w}\sin\psi_{k+1}\right]\right)\right)}^{2}+\\ {\left(A_{y,j}-\left(p_{y,k+1}+\left[t_{x,w}\sin\psi_{k+1}+t_{y,w}\cos\psi_{k+1}\right]\right)\right)}^{2}+\\ {\left(A_{z,j}-\left(p_{z,k+1}+t_{z,w}\right)\right)}^{2} \end{array}}\\ \; & =\left[\begin{array}{c} d_{jw} \end{array}\right] \end{array}$$

The set of measurement sigma points was denoted as $Z_{k+1|k}$. If the measurement used odometry, the output shape should be $n_{x}\times n_{\alpha }$, and if it was UWB range data, the output should be $1\times n_{\alpha }$.

Once the measurement sigma points were collected, the weighted predicted measurement means ($z_{k+1|k}$), and the predicted measurement covariance ($S_{k+1|k}$) were calculated, similar to how the mean and the covariance from the predicted state sigma points were calculated. The weights were defined in Section 2.5.2.
(37)$$z_{k+1|k}=\sum_{i=0}^{2n_{\sigma }}\omega_{i}^{m}Z_{k+1|k,i}$$

The measurement noise covariance was also defined $R=E\{w_{k}\cdot w_{k}^{T}\}$, where the vector $w_{k}$ was the measurement noise for the sensor’s state. For the odometry sensor, *R* would be defined as:(38)$$R=\mathrm{diag}\left(\left[\begin{array}{ccccccc} \sigma_{x}^{2} & \sigma_{y}^{2} & \sigma_{z}^{2} & \sigma_{v}^{2} & \sigma_{v}^{2} & \sigma_{\psi }^{2} & \sigma_{\dot{\psi }}^{2} \end{array}\right]\right)$$

While the *R* matrix for the UWB range sensor would be defined as:(39)$$R=\left[\begin{array}{c} \sigma_{d}^{2} \end{array}\right]$$

Using the defined *R* matrices, the predicted measurement covariance could be defined.
(40)$$S_{k+1|k}=\sum_{i=0}^{2n_{\sigma }}\omega_{i}^{c}\left(Z_{k+1|k,i}-z_{k+1|k}\right){\left(Z_{k+1|k,i}-z_{k+1|k}\right)}^{T}+R$$

#### 2.5.4. State Update

The actual state vector could be updated after defining the predicted state and measurement. First, the cross-correlation between the state points in state space and the measurement space ($T_{k+1|k}$) had to be computed:(41)$$T_{k+1|k}=\sum_{i=0}^{2n_{\sigma }}\omega_{i}^{c}\left(X_{k+1|k,i}-x_{k+1|k}\right)\left(Z_{k+1|k,i}-z_{k+1|k}\right)$$

From there, the Kalman gain was calculated:(42)$$K_{k+1|k}=T_{k+1|k}S_{k+1|k}^{-1}$$

The current state and covariance matrix were then updated, respectively, where $z_{k+1}$ was the currently retrieved measurement data:(43)$$\begin{array}{c} x_{k+1|k+1}=x_{k+1|k}+K_{k+1|k}\left(z_{k+1}-z_{k+1|k}\right) \end{array}$$(44)$$\begin{array}{c} P_{k+1|k+1}=P_{k+1|k}-K_{k+1|k}S_{k+1|k}K_{k+1|k}^{T} \end{array}$$

## 3. Simulation Results

ROS Melodic using Gazebo 9 was used to simulate the described environment and used a set of three objects/robots to model the world. The first object was a simple static UWB beacon represented by a solid cylinder meant to represent a generic UWB anchor, such as a tripod-mounted Decawave DWM1000 Module. This object would be given an agreed-upon location in global coordinates, such as GPS, and as such, the estimated position would be static and with high precision. Decawave claims they can support up to 6.8 Mbps data rates with frequencies of 3.5–6.5 GHz, a centimeter-level accuracy, and maintain a connection for up to a range of 290 m.

The second object was a Hector quadrotor, in Figure 8, with a UWB anchor mounted on the bottom of the drone [38]. This drone was meant to represent a moving UWB anchor. The drone would also be assumed to have been localized using GPS or cameras to represent a high-precision localization estimate. Instead of GPS, drones could use feature positioning based on public geographical information to remove the need to utilize GPS and generate high-accuracy localization [39,40]. In addition, as previously demonstrated in the Monte Carlo simulations in Section 2.3 in Figure 4 and Figure 5, even when noise is present in the system, if the time horizon is sufficiently large, the system will be able to derive its initial position parameters relatively accurately. Thus, even if the "localized nodes" contain noise information, the system will be able to provide an initial estimate of the position.

The final object was the implementation of the Clearpath Jackals robots depicted in Figure 9 and Figure 10 with two UWB tags on the top of its hood, equally spaced from the center axis of the robot with an extra centrally located UWB anchor. The tags allowed the robot to position itself, while the anchors allowed information propagation to help other robots localize themselves. In this study, it was assumed that the initial position and orientation of the Jackals were not provided. However, an initial estimate could be provided in the initial localization step to improve and reduce the initial position estimate error and thus improve the accuracy. As such, the localization of the Jackal was the main focus of the presented work.

The world consisted of two Jackals and three drones to simulate a mobile surveying endeavor, as presented in Figure 11 and Figure 12. The drones represented the mobile anchors as they flew over the trees and had GPS to help localize themselves with acceptable accuracy. The drones also helped the ground Jackals, who did not have GPS. The process proceeded as such: initially, the drones flew over the trees to achieve GPS localization in a global reference frame; once localized, the Jackals developed their initial position estimation based on the Hector drones. Once the Jackals had localized themselves, they performed their abstract task. Once the task was complete, the drones moved to another location to be surveyed, and the Jackals followed suit. When the Jackals moved to the new location, the drones were too far or obstructed to send UWB range information, so to continue the localization, the two Jackals sent their inter-robot range measurements and used their onboard odometry. Once the Jackals approached the new survey site, the UWB range measurements from the Hector drones helped correct any deviation during the offline path. The simulated noise measurements were based on the empirical data collected from the DWM1001-DEV in range increments of 1 m up to 141 m. The standard deviations of the measurements were recorded. The measurements also included scenarios when the UWB sensor was occluded by thin objects (<25 cm) and thick objects (≥25 cm). The distributions can be found in the Gazebo UWB Plugin https://github.com/AUVSL/UWB-Gazebo-Plugin/blob/master/src/UwbPlugin.cpp (accessed on 3 August 2023).

Two metrics were used to determine the accuracy of the unscented Kalman filter: the RMSE (root mean squared error) and mean absolute error (MAE). As demonstrated through the graphs in Figure 13, Figure 14 and Figure 15, the unscented Kalman filter developed for the Jackals follows the ground truth very closely with minimal RMSE values of no greater than 0.10921 m for the positioning errors; however, it could be noted that the unscented Kalman filter had a more significant difficulty estimating the velocity and the yaw rate, with larger errors of 0.24959 $\frac{m}{s}$ and 0.65856 $\frac{\mathrm{rad}}{s}$, though due to the difference in scales of the measurement values, this error was deemed to be acceptable. The MAE measurement values were relatively consistent, hovering around between 0.27 $\frac{m}{s}$ and 0.3 $\frac{\mathrm{rad}}{s}$, which informs of a possible issue of systematic bias introduced in the system where the position of the system was consistently offset from the actual target. As noticed through the x, Figure 13, and y, Figure 14, position graphs, if noticed closely, the estimated values for the x and y position, when the values seem steady, tend to lag below the ground-truth value. Further tuning of the unscented Kalman filter model’s parameters, such as the sensor noise matrix *R*, and the sigma point generation parameters α and β, could increase accuracy in the model. The MAE pose is the average position accuracy, a measure of how close, on average, the position of the robot was to the actual position of the robot using Euclidean distances $\left(\sqrt{{\left(x_{\exp}-x_{\mathrm{act}}\right)}^{2}+{\left(y_{\exp}-y_{\mathrm{act}}\right)}^{2}+{\left(z_{\exp}-z_{\mathrm{act}}\right)}^{2}}\right)$. Thus, on average, the robot was about 15 cm away from its target position, as noticed by the RMSE, which was often with errors that had lower magnitudes. The confidence error was calculated as the margin of error defined as $m=z^{*}\frac{\sigma }{\sqrt{n}}$, where $z^{*}$ is the level of confidence (1.96 for a 95% confidence interval), σ was the standard deviation of the variable, and *n* was the sample size. In Table 1, the confidence interval was calculated using a sample size of five runs.

The current literature does not provide a good reference comparison to this system. This is due to systems assuming a static environment or a static leader. However, no system exists where all anchors can move simultaneously and still remain localized.

## 4. Conclusions and Future Work

This article discussed UWB localization using both stationary and mobile nodes. Specifically, it introduced an ad hoc method to implement localization through a modified version of the AHLoS system. In this system, each node starts unlocalized or localized, and the localized nodes serve as reference points to the unlocalized nodes. Once there was enough information, the unlocalized nodes could determine a translation–rotation matrix from the relative position to the global reference frame. This was performed using a nonlinear least squares function, which also accounts for the tag sensor’s offset range measurements. The second phase of the paper transitioned to the UKF-based measurement data refinement stage, where both the globally translated odometry and the offset range measurements were fed into a CTRV motion-based UKF model to improve the long-term accuracy of the positioning system. Simulations were conducted to help provide proof of validity to the algorithms.

The current system assumes a 2D setup for the robot, which would cause inaccuracies in an inclined outdoor situation as the range estimation for the robot’s z value would not be correctly estimated, and the robot’s 3D dynamics would not be reflected. Another problem is that even though the possibility of obtaining an inaccurate initial position estimate decreases over time, there are no guarantees that the system will converge with this current system. More robust optimization systems will be explored in the future with explicit guarantees. In addition, the system still requires fine-tuning of specific constants, such as the initial P, Q, and R matrices of the UKF, the time horizon ‘T’, and the number of robots N for the initial localization, as suboptimal performance would be achieved if the system is not correctly estimated.

In the future, the UKF will extend the account for the robot’s yaw, pitch, and roll to better account for the diverse terrain and not assume that it was on flat terrain. To further improve the accuracy, additional sensors could be used, such as cameras and land-based markers to obtain a better position estimate and optical flow sensors to improve the velocity calculations, which will help further improve the UKF results. In addition, the development of the initial position algorithm to account for a wider variety of scenarios and improve the overall accuracy will be explored. Furthermore, a more robust adaptive Kalman filter approach for UWB sensors could be used to account for multipath and non-line-of-sight (NLOS) measurement error, which will be explored to account for a more extensive range of terrain and reduce overall position error. Finally, the current results of the paper were demonstrated in a simulated environment. Future work will be performed to demonstrate the algorithm’s feasibility in the real world.

## 5. Installation

The code for this repository for the simulated Gazebo environment can be found at https://github.com/AUVSL/UWB-Jackal-World (accessed on 18 September 2023), and the code for the UKF can be found at https://github.com/AUVSL/UWB-Localization (accessed on 26 January 2024). Both repositories contain READMEs with instructions to install/run the programs/environments. Annotated video demonstrations of the algorithms in question are located here: https://youtu.be/UbNkR3y-_S0 (accessed on 21 May 2021) and https://youtu.be/S6px8JHvk-I (accessed on 22 April 2021).

## Figures and Tables

**Figure 1 sensors-24-01154-f001:**
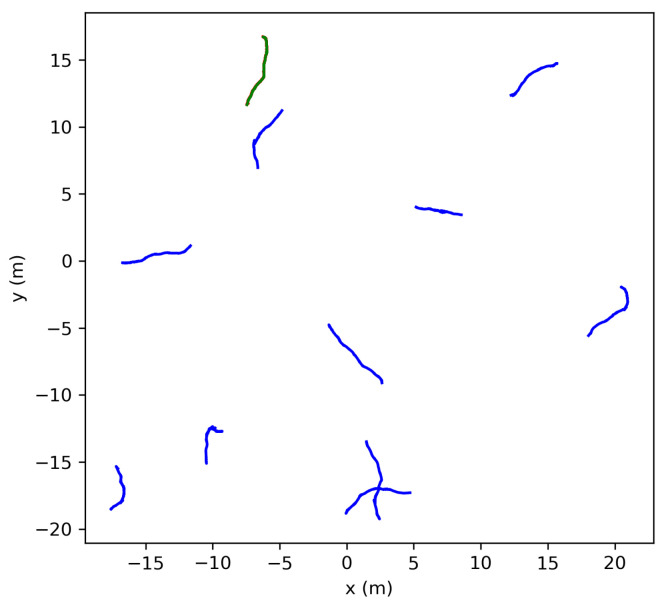
Monte Carlo simulation example with ‘T’ = 10 and ‘N’ = 10 (Green robot is the target robot).

**Figure 2 sensors-24-01154-f002:**
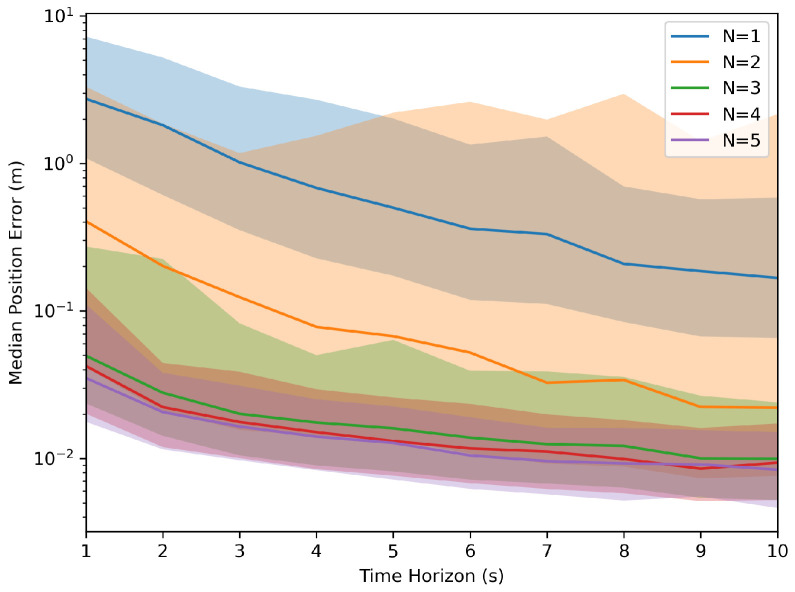
Monte Carlo simulation of the Euclidean distance error.

**Figure 3 sensors-24-01154-f003:**
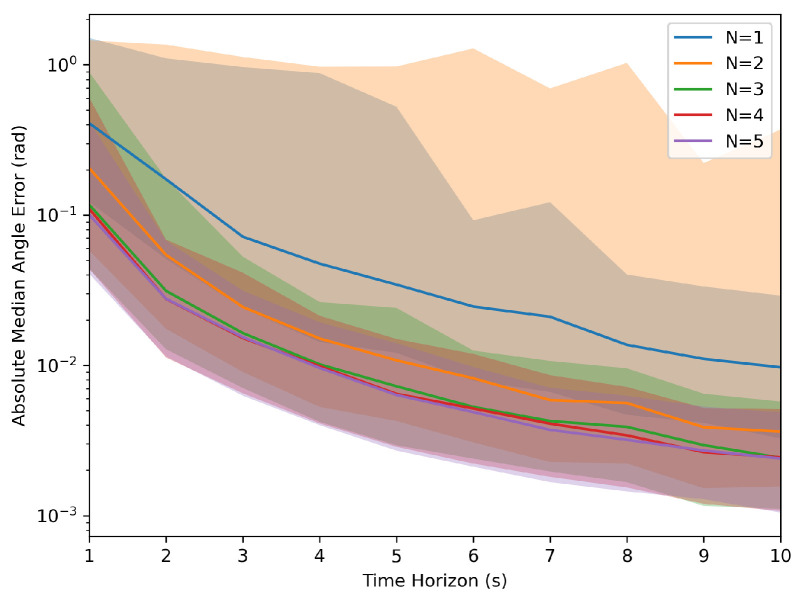
Monte Carlo simulation of the absolute angle error.

**Figure 4 sensors-24-01154-f004:**
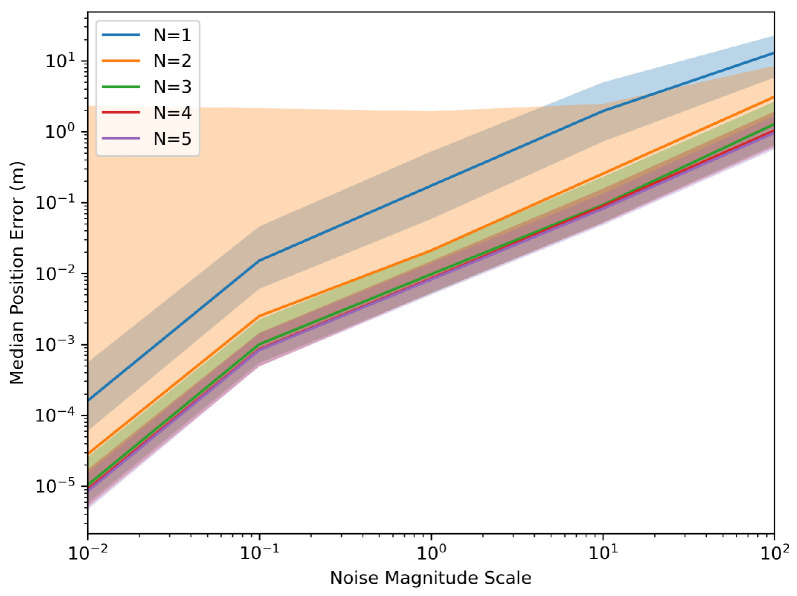
Monte Carlo of Euclidean distance error with varying noise scale.

**Figure 5 sensors-24-01154-f005:**
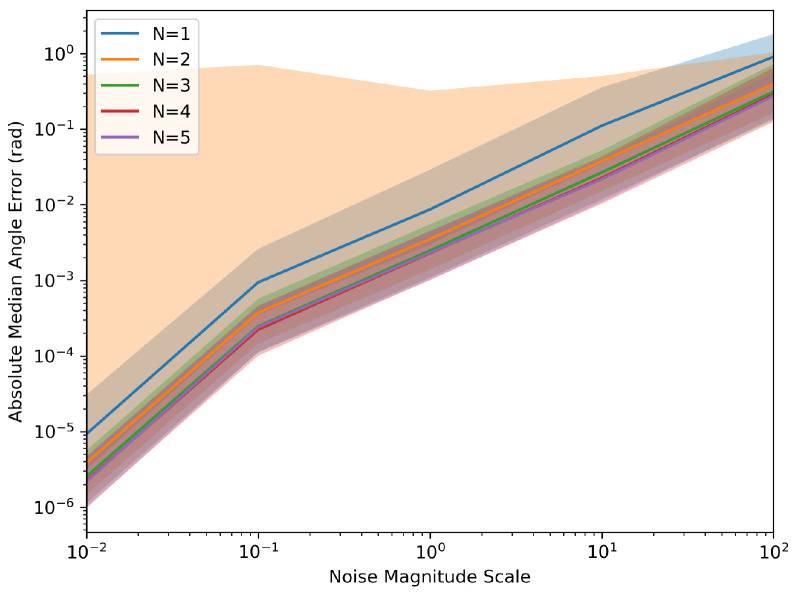
Monte Carlo of angle error with varying noise scale.

**Figure 6 sensors-24-01154-f006:**
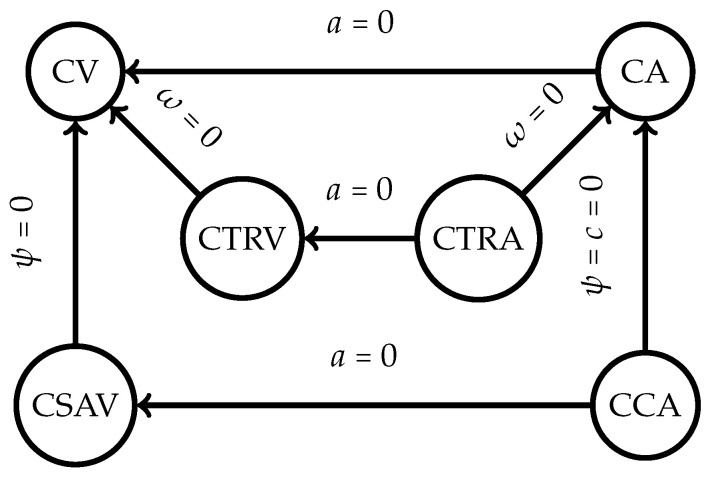
Motion model hierarchy [36].

**Figure 7 sensors-24-01154-f007:**
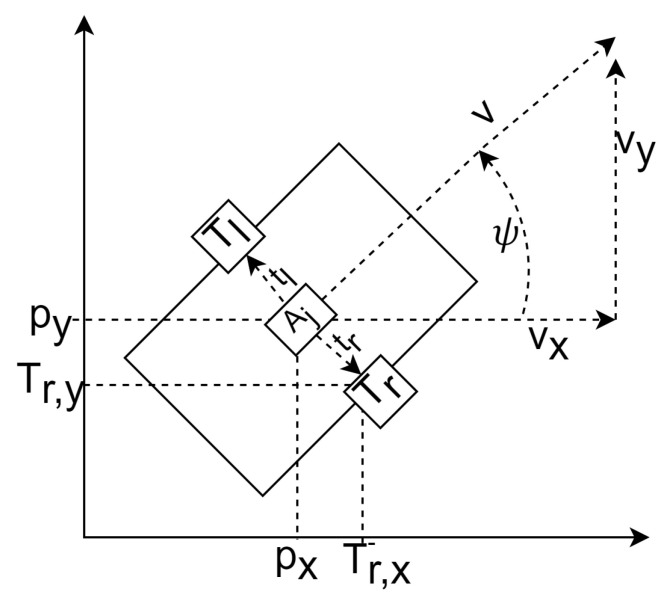
Mobile robot motion state.

**Figure 8 sensors-24-01154-f008:**
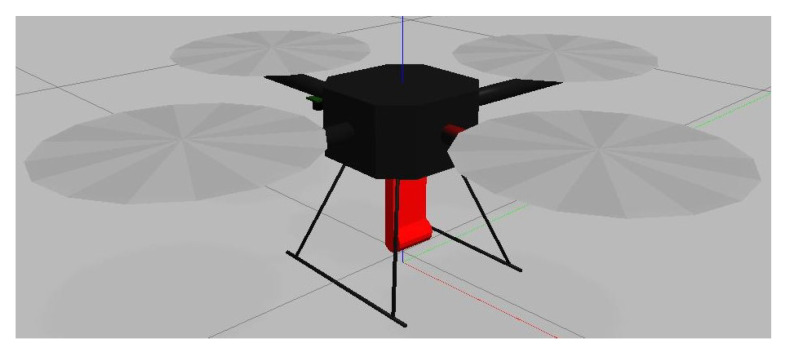
Hector drone.

**Figure 9 sensors-24-01154-f009:**
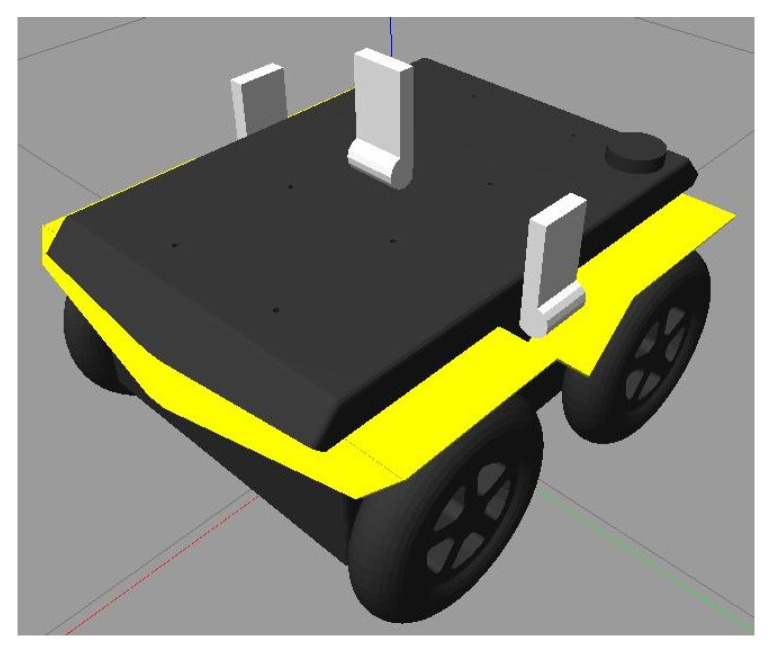
Simulated Jackal robot with UWB tags (left and right) and an anchor (middle).

**Figure 10 sensors-24-01154-f010:**
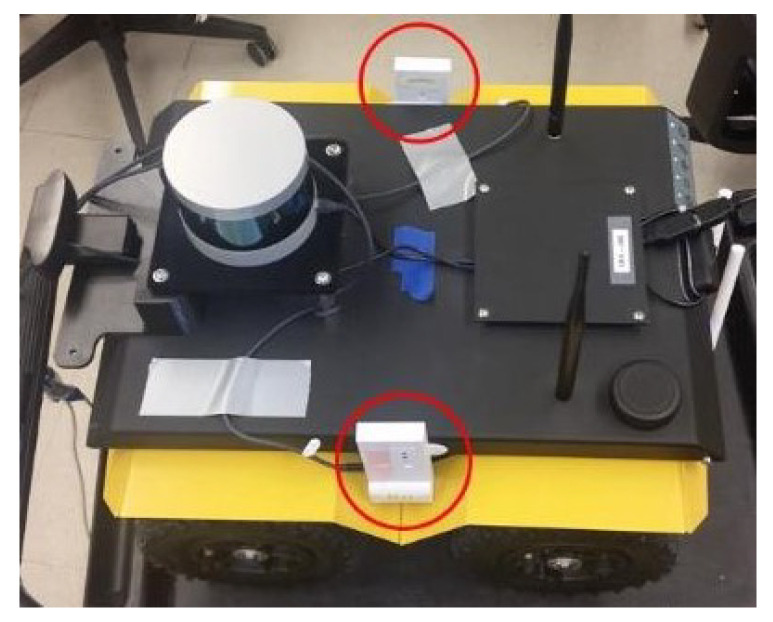
Jackal ground robot with DWMs mounted on the left and right sides [41].

**Figure 11 sensors-24-01154-f011:**
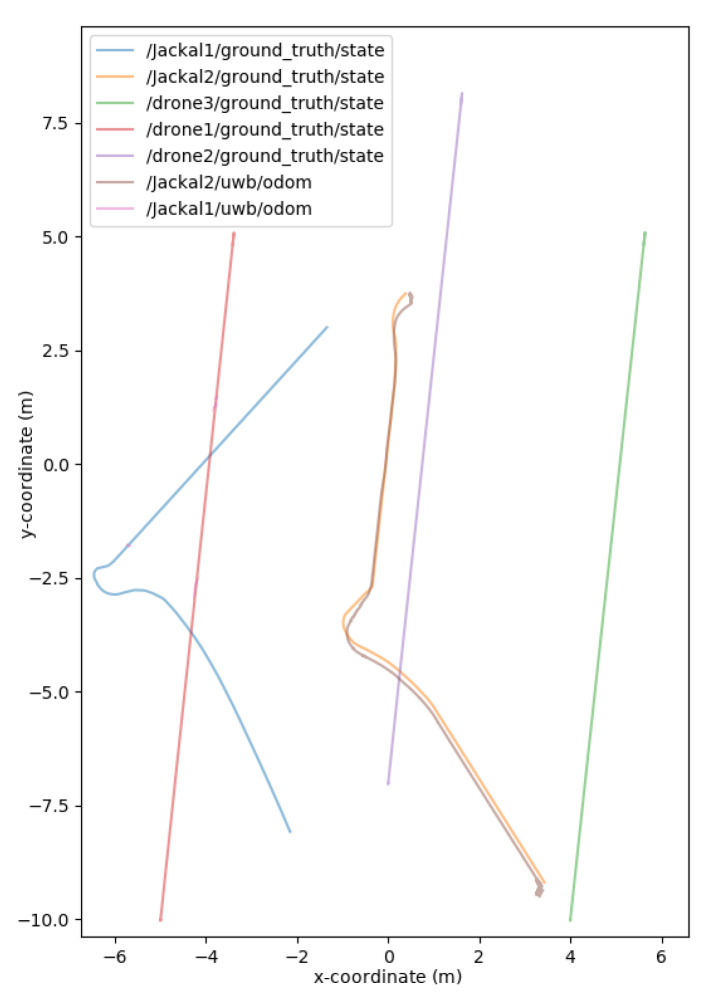
Jackal and drone position plotting.

**Figure 12 sensors-24-01154-f012:**
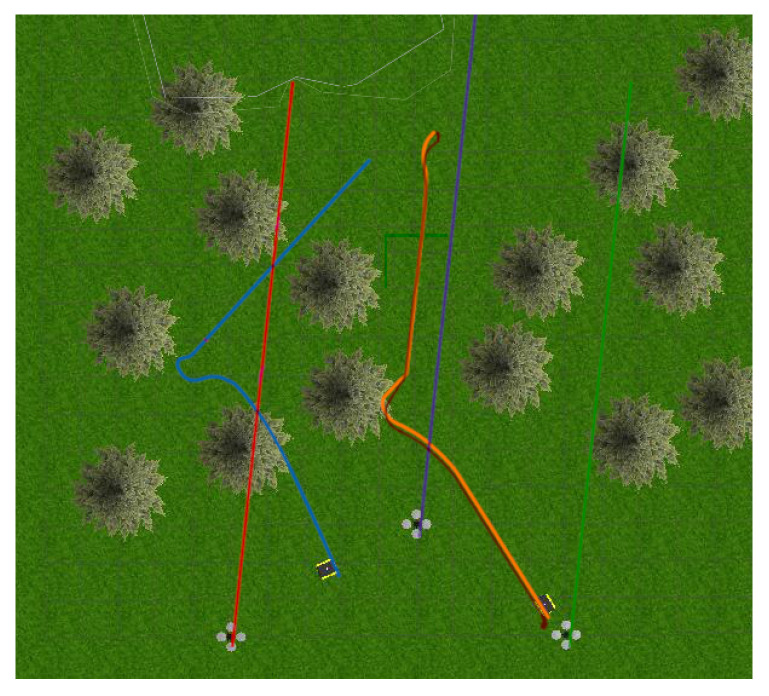
Gazebo environment position overlay.

**Figure 13 sensors-24-01154-f013:**
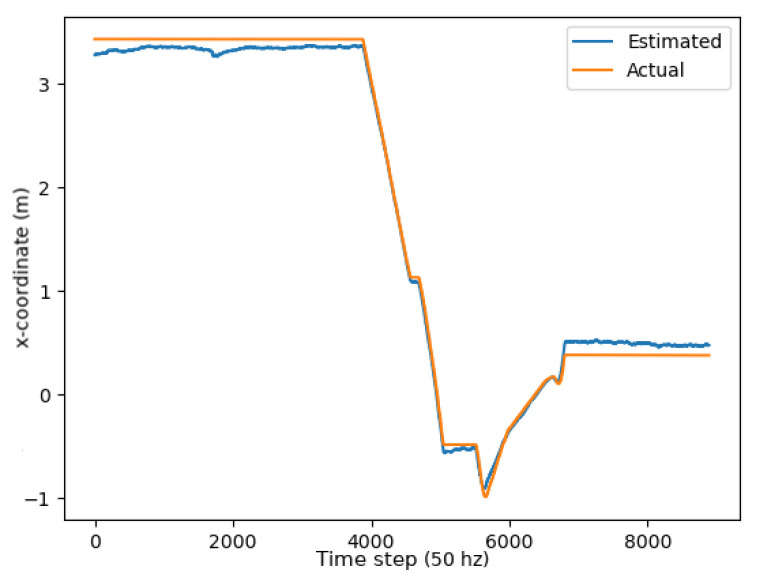
X position plot.

**Figure 14 sensors-24-01154-f014:**
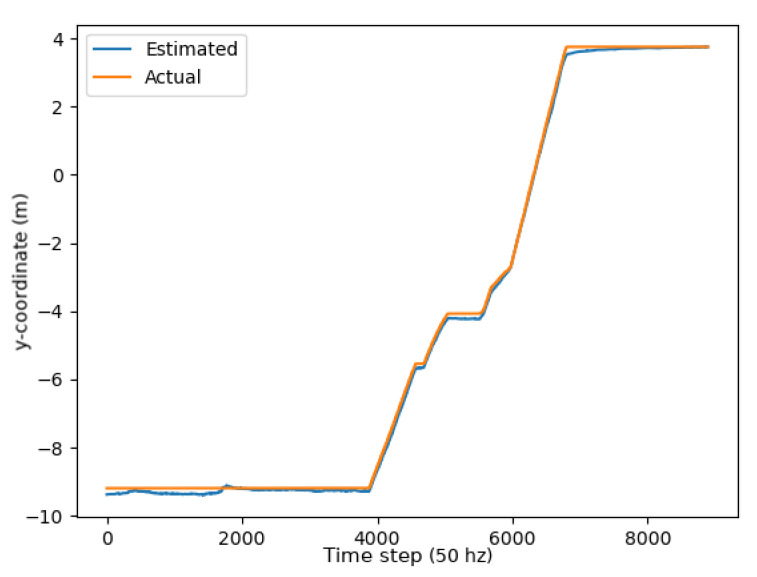
Y position plot.

**Figure 15 sensors-24-01154-f015:**
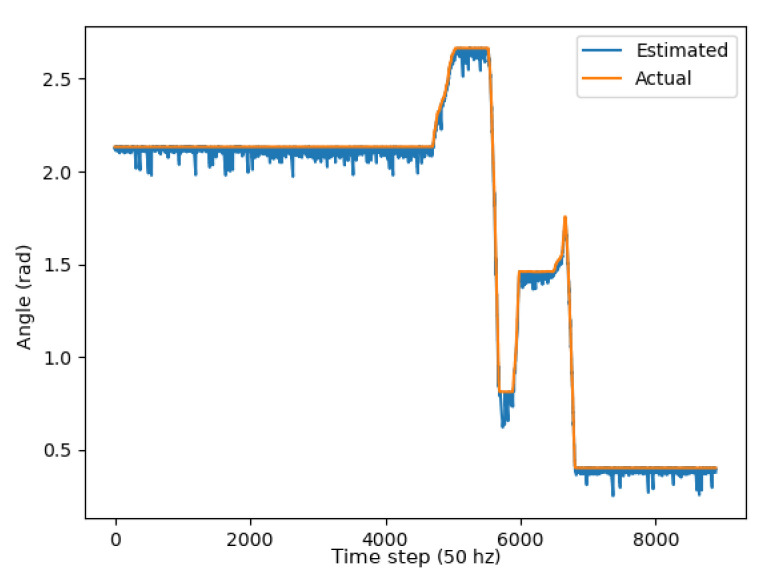
ψ plot.

**Table 1 sensors-24-01154-t001:** Drone and mobile Jackal simulation results with a 95% confidence interval.

Gazebo Worlds			
**Metrics**		**Metrics**	
RMSE x	0.0867 m ± 0.024 m	MAE x	0.2812 m ± 0.028 m
RMSE y	0.1092 m ± 0.025 m	MAE y	0.3074 m ± 0.03 m
RMSE z	0.0784 m ± 0.001 m	MAE z	0.2789 m ± 0.001 m
RMSE v	0.2496 $\frac{m}{s}$ ± 0.269 $\frac{m}{s}$	MAE v	0.3481 $\frac{m}{s}$ ± 0.14 $\frac{m}{s}$
RMSE ψ	0.0347 rad ± 0.077 rad	MAE ψ	0.1498 rad ± 0.027 rad
RMSE $\dot{\psi }$	0.6586 rad ± 8.203 rad	MAE $\dot{\psi }$	0.8114 rad ± 0.809 rad
RMSE pose	0.02559 m ± 0.01 m	MAE pose	0.15617 m ± 0.022 m

## Data Availability

Data are contained within the article.

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
