# Peer review of "Ad Hoc Mesh Network Localization Using Ultra-Wideband for Mobile Robotics"

_sensors, 2024, doi:10.3390/s24041154_

Round 1

Reviewer 1 Report

Comments and Suggestions for Authors

1.       At line 197: The calculated global transformation matrices of the UGV were defined as the transformation “from the center of the robot’s relative origin point”.
I have limited comprehension of the intended meaning of this sentence

 2. At line 198: The starting position of the odometry was assumed to be (0, 0, 0) with 0 heading, linear and angular velocity, “to the robots’ global position (χ0,0)”.

 The description is not quite clear. Please revise it.

3.       It is recommended to avoid placing images within a paragraph, such as Figure 1; instead, consider positioning them between two paragraphs to enhance readability.

4. At lines 270-272, please specify the distribution of the variable 'noise' and elucidate the significance of numerical values such as 0.5 cm, 2 cm, and 1/1000 radians.

This includes clarifying whether these values represent standard deviations or maximum errors in the context of the code.

5.  In Figures 2, 3, and 4, does the light-shaded region represent the error range? If so, why is the maximum error at N=2 often larger than at N=1? Additionally, why is the central error (medium error) for N=2 noticeably asymmetric within the error range?

6.    In Equation 11, Please clearly define βj

 7.      At lines 210-211, should Ak and Tw respectively denote the "positions" of the anchor and tag? If they represent the physical locations of the devices themselves, they should not be used in computations.

Comments on the Quality of English Language

English should be thoroughly revised and edited.

Reviewer 2 Report

Comments and Suggestions for Authors

Manuscript number:sensors-2839897

Manuscript Title: Ad-Hoc Mesh Network Localization using Ultra-Wideband for Mobile Robotics

Reviewer:

Comments to the Author

In the manuscript, UWB localization using both stationary and mobile nodes has been discussed. Specifically, it introduced an Ad-Hoc method to implement localization through a modified version of the AHLoS system. In the system, the localized nodes serve as reference points to the unlocalized nodes. Once there was enough information, the unlocalized nodes could determine a translation-rotation matrix from the relative position to the global reference frame. This was done using a non-linear least squares function which also accounts for the tag sensor’s offset range measurements.

This work is interesting. Yet, the following are comments of the manuscript for further improvements.

1. How about the test results of the proposed method?

2. Figure 13 and Figure 14 give the X and Y position plot. Why the different between the estimated results and actual data for X position more than that for Y position?

3. How to simulated the method in the manuscript? The details should be given in the reviewed manuscript.  

Reviewer 3 Report

Comments and Suggestions for Authors

This work aims to demonstrate the localization of bicycle-modeled robots in a non-static environment through a mesh network of mobile, stationary robots and ultra-wideband sensors. There are few issues in the paper and hence I strongly recommend below corrections:

1. There are few mistakes, typo errors from the beginning till the end. Please read carefully and submit the revised version. For ex, the line "There is little research on Ad-Hock UWB-enabled localization systems with mobile and stationary nodes" Is it right?

2. Section 1 and 2 are fine.

3. In Monte Carlo simulation, the type of random distribution or the value fixation to be clearly mentioned.

4. How many runs the simulation was conducted?

5. The confidence interval is not mentioned.

6. The deviation in the estimated and the accuracy for different scenarios to be tabulated.

7. The authors may not include the result graphs in the conclusion. Bring back those results to previous section and write a simple paragraph to conclude the work and highlight the future work also.

8. References are adequate and fine.

Comments on the Quality of English Language

Minor editing of English language required

Reviewer 4 Report

Comments and Suggestions for Authors

This article explores high-accuracy GPS-denied Ad-Hoc localization, focusing on UWB-enabled systems for mobile and stationary nodes. The study demonstrates bicycle-modeled robot localization in a dynamic environment using a mesh network and ultra-wideband sensors. Initial localization employs AHLos, refined with a UKF based on a CTRV model. The robot shares its estimated position with neighbors, creating a wave-like cycle for mutual improvement, transforming the network into a mobile Ad-Hoc localization network.

Following are my comments:

-          The mentioned contributions should be more rigorous. Authors should justify the novelty of their works.

-          In this work, only simulation results are presented. Although the simulation results are good and well presented, this research will be more reliable with such real experiments. I think that the real validation can promote the presented work.

Round 2

Reviewer 1 Report

Comments and Suggestions for Authors

All my questions were answered. I have no comments about this submission.

Reviewer 3 Report

Comments and Suggestions for Authors

The authors have carried out all suitable corrections suggested by the reviewers and they have improved the paper well. Hence the paper shall be accepted in present form.

Reviewer 4 Report

Comments and Suggestions for Authors

This paper can be accepted in its current form, No further comments.